# Anthropometric Measurements for Predicting Low Appendicular Lean Mass Index for the Diagnosis of Sarcopenia: A Machine Learning Model

**DOI:** 10.3390/jfmk10030276

**Published:** 2025-07-17

**Authors:** Ana M. González-Martin, Edgar Samid Limón-Villegas, Zyanya Reyes-Castillo, Francisco Esparza-Ros, Luis Alexis Hernández-Palma, Minerva Saraí Santillán-Rivera, Carlos Abraham Herrera-Amante, César Octavio Ramos-García, Nicoletta Righini

**Affiliations:** 1Instituto de Investigaciones en Comportamiento Alimentario y Nutrición (IICAN), Universidad de Guadalajara, Ciudad Guzmán 49000, Jalisco, Mexico; samid.limon@cusur.udg.mx (E.S.L.-V.); zyanya.reyes@cusur.udg.mx (Z.R.-C.); alexis.hernandez@cusur.udg.mx (L.A.H.-P.); 2Injury Prevention in Sport Research Group, Universidad Católica San Antonio de Murcia (UCAM), 30107 Murcia, Spain; fesparza@ucam.edu; 3Departamento de Ciencias Clínicas, Universidad de Guadalajara, Ciudad Guzmán 49000, Jalisco, Mexico; minerva.santillan@cusur.udg.mx; 4Nutritional Assessment and Nutritional Care Laboratory (LECEN), Division of Health Sciences, Centro Universitario de Tonalá, Universidad de Guadalajara, Tonalá 45425, Jalisco, Mexico; carlos.amante@academicos.udg.mx (C.A.H.-A.); octavio.ramos@academicos.udg.mx (C.O.R.-G.); 5Ibero-American Network of Researchers in Applied Anthropometry (RIBA2), 04120 Almería, Spain

**Keywords:** anthropometry, aged, artificial intelligence, Mexico, muscles, sarcopenia

## Abstract

**Background**: Sarcopenia is a progressive muscle disease that compromises mobility and quality of life in older adults. Although dual-energy X-ray absorptiometry (DXA) is the standard for assessing Appendicular Lean Mass Index (ALMI), it is costly and often inaccessible. This study aims to develop machine learning models using anthropometric measurements to predict low ALMI for the diagnosis of sarcopenia. **Methods**: A cross-sectional study was conducted on 183 Mexican adults (67.2% women and 32.8% men, ≥60 years old). ALMI was measured using DXA, and anthropometric data were collected following the International Society for the Advancement of Kinanthropometry (ISAK) protocols. Predictive models were developed using Logistic Regression (LR), Decision Trees (DTs), Random Forests (RFs), Artificial Neural Networks (ANNs), and LASSO regression. The dataset was split into training (70%) and testing (30%) sets. Model performance was evaluated using classification performance metrics and the area under the ROC curve (AUC). **Results**: ALMI indicated strong correlations with BMI, corrected calf girth, and arm relaxed girth. Among models, DT achieved the best performance in females (AUC = 0.84), and ANN indicated the highest AUC in males (0.92). Regarding the prediction of low ALMI, specificity values were highest in DT for females (100%), while RF performed best in males (92%). The key predictive variables varied depending on sex, with BMI and calf girth being the most relevant for females and arm girth for males. **Conclusions**: Anthropometry combined with machine learning provides an accurate, low-cost approach for identifying low ALMI in older adults. This method could facilitate sarcopenia screening in clinical settings with limited access to advanced diagnostic tools.

## 1. Introduction

Currently, there has been a sustained increase in the global elderly population [1]. In some countries, for the first time, the number of adults over the age of 65 has surpassed that of children under five years old, and projections estimate that by 2050, the number of individuals in this age group will equal that of those under 14 years old [2]. This trend is significant because aging is associated with a higher prevalence of chronic diseases, which increases healthcare costs for both individuals and their families while also placing a greater financial burden on public health systems [3,4,5]. As a result, some have even proposed aging itself as a disease specific to this stage of life [3].

Sarcopenia is a muscular disease characterized and diagnosed by low muscle mass, decreased muscle strength, and poor physical performance, all of which negatively affect mobility and quality of life in older adults [6,7]. Proper identification and interpretation of the condition are essential for its prevention and treatment [8]. Several international consensus groups have established guidelines and methods for measuring these diagnostic criteria [6,9,10]; however, some of these methods—particularly those used to assess Appendicular Lean Mass Index (ALMI)—are not accessible in all clinical settings. For instance, dual-energy X-ray absorptiometry (DXA), while considered a standard reference, requires expensive and often unavailable equipment, making sarcopenia screening difficult in many regions [6,9,11]. Moreover, in “The Position Statements of the Sarcopenia Definition and Outcomes Consortium,” published in 2020, the clinical utility of DXA for muscle mass estimation was questioned. Although muscle mass remains a central diagnostic criterion, panelists expressed concerns about relying on DXA-derived lean mass, citing evidence that it is not consistently associated with adverse health outcomes in community-dwelling older adults, even when adjusted for body size. They suggested that more accurate alternatives, such as D3-creatine dilution, may show stronger associations with clinical outcomes and could be prioritized in future practice [12].

Another challenge in diagnosing sarcopenia is the use of population-specific cut-off points established by various consensus groups, such as the Asian Working Group for Sarcopenia (AWGS) and the European Working Group on Sarcopenia in Older People (EWGSOP), the latter being the most widely used [6,9,13]. This is important because studies have shown variations in cut-off values across different populations [14,15]. Accordingly, several countries are working to develop diagnostic tools and validated cut-off points tailored to their populations [14,15,16,17,18,19,20].

Additionally, sarcopenia prevalence estimates can vary widely—between 4.6% and 41%—depending on the diagnostic criteria used. This may be due to significant differences in body composition across ethnicities [21]. The existence of multiple diagnostic methods and cut-off values leads to low concordance and may result in underdiagnosis or overdiagnosis [12,22,23]. Therefore, it is recommended to estimate prevalence using ethnicity-specific cut-off points [22].

Accurately measuring lean mass—both in terms of quantity and quality—remains a challenge, as the most reliable assessment tools are costly and often inaccessible [11]. This has led to increasing efforts to explore alternative methods for estimating sarcopenia-related variables, including the use of anthropometry to predict lean mass through formulas or girth measurements, particularly of the leg, which is the only anthropometric measure currently validated for case detection [6,11,24,25]. Quadriceps girth has also emerged as a promising technique to estimate both the quantity and quality of muscle mass [11]. Anthropometry offers key information on health status and is a cost-effective, non-invasive, and universally applicable method for measuring body mass, proportions, and composition [24]. However, further research is needed to evaluate and compare lean mass estimation methods, particularly those adapted to specific ethnic groups. Differences in body composition between ethnic groups may explain the need for such adaptations. For example, Alemán-Mateo et al. [21] found that Mexican individuals had higher fat mass and lower lean mass compared to African American and Caucasian individuals in the United States. Moreover, body composition variability has been documented even among different regions within Mexico, further complicating the establishment of standardized cut-off points [26]. These variations likely contribute to inconsistencies in diagnostic outcomes, emphasizing the importance of conducting validation procedures specific to each demographic group [11]. In this context, research focused on older Mexican people remains limited, particularly regarding body composition estimation methods that are both validated for this demographic and practical for use in clinical settings. This limitation is primarily due to restricted access to advanced validation tools, such as DXA, as well as the heterogeneity of existing studies and a predominant focus on younger and general adult populations. These factors present significant challenges for standardization and meaningful comparisons across studies. Therefore, additional studies are necessary to evaluate and compare various muscle mass assessment methods, tailored to the specific characteristics of this population.

The use of artificial intelligence (AI) tools—particularly machine learning (ML)—has become increasingly prevalent in health-related predictive modeling [27]. One of ML’s key advantages in healthcare is the ability to automate data processing, thereby improving the efficiency of assessment, diagnosis, treatment, and monitoring [27,28,29]. AI algorithms such as Support Vector Machines (SVMs), Decision Trees (DTs), and Random Forests (RFs), among others, have proven useful for diagnosing sarcopenia and osteoporosis, offering predictive variables adapted to specific populations [14,29,30]. These methods have also been applied in models for diagnosing cardiovascular diseases, overweight and obesity, atrial fibrillation, endometrial lesions, and more [14,29,30,31,32,33,34,35,36]. A notable advantage of predictive risk models in clinical practice is their ability to provide more individualized risk assessments, thus enhancing the efficacy of interventions [35].

Considering the above, there is a growing need to develop accessible, accurate, and adaptable computational models capable of reliably estimating ALMI in resource-limited clinical settings. This study proposes the design of a predictive model based on anthropometric variables to detect low ALMI in Mexican adults. Such a tool could facilitate early sarcopenia detection and improve clinical decision-making, particularly in populations with limited access to advanced technologies. Furthermore, it could contribute to improving intervention methods and reducing the healthcare costs associated with functional decline in old age.

## 2. Materials and Methods

### 2.1. Study Design and Participants

Participants were selected using a non-probabilistic purposive sampling method, and participation was entirely voluntary. The minimum size was calculated to be 96 older adults from the state of Jalisco, including both males and females. The sample size was estimated using the following standard formula for descriptive studies, assuming a 95% confidence level:
$$n=\frac{{\left(Z_{\alpha /2}\right)}^{2}(p)(q)}{d^{2}}$$ where *Z*_α/2_ = 1.96, *q* = 1 − *p*, estimated prevalence (10%): *p* = 0.10, and standard deviation of the prevalence (6%): *d* = 0.06.

This study adhered to the Strengthening Reporting of Observational Studies in Epidemiology (STROBE) guidelines [37]. A cross-sectional design was implemented with a sample of older Mexican adults aged 60 years and above, of both sexes. The recruitment strategy involved visits to institutions that gather older adults, such as health centers and recreational facilities. This study was named “Sarcopenia in Older Adults from Jalisco (SAMJ)”. Individuals with limb loss, inability to move independently, or the presence of edema were excluded.

### 2.2. Instruments

#### 2.2.1. Body Composition Assessment Using DXA

A Lunar iDXA densitometer (General Electric) was used. This equipment emits X-rays at two different energy levels, which are attenuated as they pass through tissues depending on their density. Using these attenuation coefficients, an R value is obtained, enabling the calculation of total and regional lean mass, fat mass, bone mass, and fat-free mass. From these data, Appendicular Lean Mass (ALM) was calculated for sarcopenia diagnosis.

The DXA evaluation process included the following three phases:Participant preparation: participants were asked to avoid intense physical activity.Positioning: participants were set in a supine position on the scanning table, free of external metal objects. The limbs were positioned alongside the body, palms facing downward, feet in a neutral or slightly inward position, and face facing upward in a neutral position.Post-processing: scans were performed by trained personnel following the manufacturer’s instructions [38].

#### 2.2.2. Anthropometry

All measurements were taken on the participants’ right side by a Level 3 ISAK-certified anthropometrist (A.M.G.-M.), following the standardized protocol of the International Society for the Advancement of Kinanthropometry (ISAK) [39]. Anthropometric measurements were obtained using a portable stable stadiometer (Seca 217, Seca GmbH & Co. KG, Hamburg, Germany), a wide platform digital weighing scale (model XL-700, Detecto, Webb City, Missouri, USA), a steel anthropometric tape (Lufkin W606PM, Lufkin, Missouri City, Texas, USA), and a skinfold caliper (Baty International, Harpenden, UK).

Before any measurement, the following anatomical landmarks were identified and marked with a dermatographic pencil:Acromiale: upper border of the most lateral part of the acromion.Radiale: proximal–lateral border of the radial head.Mid acromiale–radiale: midpoint between the acromiale and radiale marks.Triceps skinfold: posterior midline of the upper arm, aligned with the mid acromiale–radiale point.Calf skinfold: medial aspect of the leg at its maximum girth.

Measurements taken:Body mass: measured using a calibrated digital scale, with the participants in anthropometric position (standing, feet shoulder-width apart, arms relaxed at the sides).Stretch stature: measured with a stadiometer, participants barefoot and upright, feet together, head in the Frankfurt plane.Triceps skinfold (TSF): taken with a skinfold caliper, vertically at the mid acromiale–radiale point, parallel to the arm’s longitudinal axis.Calf skinfold (CSF): participant standing with the right foot on the anthropometric box and right knee flexed at 90°. The skinfold was measured vertically at the designated point on the medial leg.Arm Relaxed Girth (ARG): measured at the mid acromiale–radiale point with a non-elastic tape, arm hanging naturally.Arm Flexed and Tensed Girth (AFTG): arm flexed at 90° in front of the body. Participants were asked to contract the biceps maximally, and the girth was measured at the point of greatest muscle prominence.Forearm Girth (FG): with the forearm slightly flexed and in a supine position, the maximum girth was measured.Calf Girth (CG): with the participants standing on the anthropometric box, measured at the level of the leg skinfold point.

Corrected Girths (CoGs) were calculated using the following formula: CoG = Limb girth (cm) − (Skinfold (cm) × π) [40,41,42]. Each measurement was taken twice, and if the difference exceeded 5%, a third measurement was performed. The average of the closest two was recorded. The equipment was calibrated daily, and all measurements were taken by the same evaluator (A.M.G.-M.) to ensure reliability.

### 2.3. Procedures

This study was carried out in collaboration with the Laboratorio de Evaluación y Cuidado del Estado Nutricio (LECEN) at the Universidad de Guadalajara (CUTonalá) and the Centro de Atención Integral al Adulto Mayor (CAIAM) of the Sistema Nacional para el Desarrollo Integral de la Familia (DIF) Tonalá and Tlaquepaque. Initial recruitment occurred at the CAIAM in Guadalajara, where older adults attending activities were invited to participate. An agreement was made with the institution’s director to arrange transportation to CUTonalá for anthropometric assessments by scheduled appointments.

Evaluations were conducted twice a week, with 10 to 17 participants assessed per session. Upon arrival at LECEN (10:00 am), participants were welcomed in a large classroom adjacent to the laboratory where measurements were taken. Participants were informed about this study and its implications, and informed consent was obtained. A patient rotation system was then organized. Participants first attended the anthropometry station, followed by the DXA scan, and then proceeded to other assessment stations (results not discussed in this article). Once the first participant had completed the full circuit, the next participant was evaluated, continuing in this manner until all individuals who attended that day had been assessed. While waiting, participants engaged in games and had access to snacks, coffee, tea, and water. Anthropometric and body composition evaluations were carried out following standardized protocols. For DXA scans, participants wore a medical gown and avoided underwear containing metal. For anthropometric measurements, they wore lightweight pants that could be rolled up to the knee and a sleeveless shirt. Minimal clothing was not required, prioritizing participant comfort.

### 2.4. Data Analysis

Microsoft Excel and RStudio (version 2024.09.1+394) were used for data analysis. Cut-off points for ALM were determined using the 20th percentile [43,44,45,46]. Normality of the data was assessed via histograms, Q-Q plots, and the Kolmogorov–Smirnov test. Pearson or Spearman correlations were used based on data distribution to assess relationships between DXA-derived ALM and anthropometric variables.

Subsequently, artificial intelligence tools were implemented using machine learning models, including Random Forests (RFs), Logistic Regression (LR), Decision Trees (DTs), Artificial Neural Networks (ANNs), and Least Absolute Shrinkage and Selection Operator (LASSO) regression. Models were trained on 70% of the dataset and tested on the remaining 30%. The target variable was binary (“normal” vs. “low” ALMI) and was predicted using anthropometric variables. Full details of the analysis can be reviewed in the R Markdown available in the Appendix A.

### 2.5. Ethical and Biosafety Considerations

The SAMJ research project was approved by the Research and Graduate Studies Committee (registration number SAC/CIP/DOAN/027/2023) and by the Ethics Committee (registration CEI/77/2023, approved on 8 March 2025) of the Centro Universitario del Sur, Universidad de Guadalajara. This study followed the ethical principles outlined in the Declaration of Helsinki [47] and the Reglamento de la Ley General de Salud en Investigación en Salud (1987) of Mexico [48]. Informed consent and/or assent were obtained in accordance with Articles 20–22 of the aforementioned regulation.

Participation was entirely voluntary. Recruitment was conducted by health and research staff through direct invitations and informational posters at participating institutions. As a benefit, participants received a health report including body composition, diet, and physical performance assessment, valued at approximately 13,000 Mexican pesos. Results were delivered in person or via the participant’s preferred method (phone call, WhatsApp, text, or email).

## 3. Results

### 3.1. General and Descriptive Result

The sample consisted of 183 older adults, 67.2% women and 32.8% men. Most participants were either married (41%) or widowed (35%), lived with their family (67.8%), did not smoke (90.2%), did not consume alcohol (84.7%), did not have insulin resistance (85.8%), and had no diagnosed depression (80.9%) (see Section A.1). Sex-based comparisons of the variables are shown in Table 1 and Table 2. Significant sex differences were found in most variables, except for BMI and age.

#### Sarcopenia

Table 3 presents the cut-off points derived from the SAMJ study sample, which were used to identify and categorize ALMI levels as low or normal.

### 3.2. Body Composition Values Related to Anthropometry

Table 4 shows the correlations between anthropometric measurements and body composition variables assessed via DXA. The following variables indicated strong (r > 0.7; [49]) and statistically significant positive correlations: (a) body mass with ALM and lean mass in both arms and legs, including ALMI; (b) BMI, ARG, triceps, and leg skinfolds with fat mass values; (c) forearm girth with lean mass values, except for the ALM/BMI ratio; (d) calf girth with ALMI; and (e) corrected calf and arm girths with ALM and lean mass in the arms and legs (Figure 1).

### 3.3. Predictive Models for Appendicular Lean Mass Index (ALMI) Using Anthropometry

ALM is a key diagnostic criterion for sarcopenia, yet one of the most difficult variables to assess in clinical practice. Therefore, estimating it using anthropometric variables is a priority and also a practical alternative. In this study, the aim was to predict the binary classification of ALMI as “normal” or “low.” The following machine learning (ML) models were applied: Decision Trees (DTs), Logistic Regression Models (LR), Random Forests (RFs), Artificial Neural Networks (ANNs), and LASSO regression (LASSO). The models were evaluated using AUCs and specificity values.

#### 3.3.1. Logistic Regression Models

The predictive values of the variables included in the model were obtained (Table 5). When classifying the cases in the test process, the model achieved a classification AUC value of 0.76 for females and 0.77 for males, which means they are able to identify 76% and 77% of the cases with low ALMI in the test sample (see Section 3.3.6).

#### 3.3.2. Decision Trees

In Figure 2 and Table 6, BMI and CCG were identified as the most relevant predictors of low ALMI for women, while ARG was the most relevant for men. Cut-off points for each variable were established to detect low ALMI. Model validation through case classification tests yielded an AUC of 0.84 for women and 0.80 for men, with slightly lower performance observed in males (Table 7).

**Table 6 jfmk-10-00276-t006:** Decision Tree model for ALMI prediction by sex in older adults from the SAMJ study.

Female						Male					
Node	split	n	loss	yval	(yprob)	Node	split	n	loss	yval	(yprob)
(1)	root	87	18	1	(0.793–0.207)	(1)	root	42	8	1	(0.809–0.191)
(2)	BMI ≥ 25.83	61	1	1	(0.984–0.016) ☨	(2)	ARG ≥ 28.75	30	0	1	(0.000–0.000) ☨
(3)	BMI < 25.83	26	9	2	(0.346–0.333)	(3)	ARG < 28.75	12	4	2	(0.333–0.666) ☨
(4)	CCG ≤ 26.89	12	4	1	(0.666–0.333) ☨						
(5)	CCG < 26.89	14	1	2	(0.071–0.929) ☨						

☨ Denotes terminal node. split: splits; n: number of cases; yval: the mean response value of all observations in the training dataset; yprob: estimated probability of each class at each terminal node of the tree; BMI: Body Mass Index; CCG: Calf Corrected Girth; ARG: Arm Relaxed Girth.

#### 3.3.3. Random Forest

When the modeling was carried out, the variables with the greatest predictivity for ALMI level were identified as BMI and CG for females and CG and ARG for males (Figure 3). Table 7 shows the results of case classification during the test phase. The BAs indicated an AUC of 0.82 for females and 0.79 for males (Table 7).

#### 3.3.4. Artificial Neural Network

Figure 4 displays the ANNs, each consisting of a single hidden layer with five neurons (H1 to H5) and one output neuron (O1) responsible for predicting the binary outcome: low vs. normal ALMI. Bias nodes B1 and B2 are included to adjust activation thresholds and enhance the learning process. The connecting lines represent the synaptic weights, where thicker lines indicate stronger connections. Line color denotes the direction of the weight: black for positive and burgundy for negative weights. The distribution of weights from input to hidden neurons appears more balanced in the network shown in Figure 4b. In the female model (Figure 4a), AFTG emerges as the most influential variable, followed by ARG and BMI, indicating a greater predictive contribution from thigh-related anthropometric measures. In the male model (Figure 4b), CCG appears to be the most important predictor, followed by CG and ARG. Regarding performance, the models achieved an AUC of 0.82 for ALMI classification in females and 0.92 for males on the test dataset (Table 7).

#### 3.3.5. LASSO Regression

LASSO regression is a technique that combines coefficient shrinkage and variable selection to enhance model performance and interpretability in regression models [50]. Table 7 summarized the performance metrics of the model for ALMI classification, showing an AUC of 0.84 for females and 0.87 for males.

#### 3.3.6. Performance Metrics and Model Comparison

Table 7 shows the performance metrics of all the evaluated models for ALMI classification. For females, both the DT and LASSO models demonstrated the best predictive performance, each achieving an AUC of 0.84. In the case of males, the ANN model outperformed others with an AUC of 0.92. These results are visually represented in Figure 5 through the ROC curves. Given that the primary goal in predicting sarcopenia is to accurately identify low ALMI cases, specificity is a particularly important metric. All models for females achieved high specificity scores above 0.93, with the DT model exhibiting perfect specificity (1.00), meaning it correctly identified all true negative cases (i.e., correctly identifying all the cases with low ALMI). For male participants, specificity values were slightly lower but still clinically relevant. LR had the lowest specificity at 0.77, whereas the RF model performed best, reaching a specificity of 0.92 in detecting negative cases.

**Table 7 jfmk-10-00276-t007:** Classification performance metrics to predict ALMI level with anthropometric variables in older adults from the SAMJ study.

	Female					Male				
	**LR**	**DT**	**RF**	**ANN**	**LASSO**	**LR**	**DT**	**RF**	**ANN**	**LASSO**
Accuracy	0.86	0.92	0.86	0.86	0.89	0.77	0.82	0.88	0.88	0.82
Sensitivity	0.57	0.57	0.57	0.57	0.57	0.75	0.75	0.75	0.75	0.75
Specificity	0.93	1.00	0.93	0.93	0.97	0.77	0.85	0.92	0.85	0.85
Precision	0.67	1.00	0.67	0.67	0.80	0.50	0.60	0.75	0.60	0.60
AUC	0.76	0.84	0.82	0.82	0.84	0.77	0.80	0.79	0.92	0.87

LR: Logistic Regression; DT: Decision Tree; RF: Random Forest; ANN: Artificial Neural Network; LASSO: LASSO regression.

## 4. Discussion

The results of ALMI level modeling are presented based on the performance of five machine learning approaches: DT, LR, RF, ANN, and LASSO regression. In DT models, the most important predictive variables were BMI and calf girth (CCG) for females and arm girth (ARG) for males. The DT model achieved the highest performance in females, with an AUC of 0.84. For males, the ANN model demonstrated the strongest predictive capability, achieving a test accuracy of 0.92 with low error rates. Overall, BMI emerged as a particularly influential predictor in models for women. In terms of specificity for detecting low ALMI, the DT model achieved perfect classification (100%) in females, showing high accuracy in identifying true negatives while maintaining good detection of low ALMI cases; in males, the RF model performed best, with a specificity of 93%.

The results of this modeling suggest that DT, RF, and ANN are promising tools for predicting low ALMI using anthropometry. Further training and validation could generate normative reference values useful in clinical practice. These ML models are commonly used for case classification, offering strong potential in clinical settings—particularly for distinguishing between the presence and absence of disease—by capturing complex, non-linear, and heterogeneous relationships among variables, even when the underlying physiological mechanisms are not fully understood due to biological complexity or pathological variability [51].

Among these, DTs are especially well-suited for clinical applications due to their intuitive structure, high classification accuracy, ease of validation by clinical experts, and user-friendly, human-readable format. They classify new instances based on patterns identified from previously labeled data, making them valuable for disease diagnosis and risk stratification. In our study, DTs enabled the identification of a small set of highly predictive features (e.g., BMI, CCG, and ARG) that could guide the design of simple, low-cost screening tools. However, DTs can be sensitive to noisy data and prone to overfitting, which may compromise generalizability—limitations that are less common in algorithms such as ANNs. While ANNs are more robust to noise, they often require longer training times and are less interpretable due to their “black box” nature [52]. RF, an ensemble method built from multiple DTs through bootstrap sampling, mitigates some of the weaknesses of single-tree models. It enhances predictive performance by reducing model variance and increasing stability [51,53]. Each tree in the forest independently generates a prediction, and the final output is determined by a majority vote (for classification) or by averaging predictions (for regression). As a non-parametric algorithm, RF is well-suited for both continuous and categorical variables and is relatively simple to tune [53]. Additional advantages include its robustness to overfitting, tolerance to outliers, and ability to compute ancillary metrics such as classification error and variable importance through permutation testing [51,53]. This capacity to quantify feature relevance, combined with its strong generalization capabilities, makes RF a valuable tool in biomedical research and clinical predictions [53]. Despite these strengths, the applicability of our models is currently limited by the absence of external validation. While internal cross-validation was applied, future research should involve independent datasets from diverse populations to evaluate reproducibility, generalizability, and key requirements for clinical implementation. In summary, both DT and RF models offer significant advantages for practical implementation in clinical contexts, given their interpretability and support for informed decision-making. However, the limited sample size and sex imbalance in our study may affect the stability and reliability of the models, particularly among male participants, underscoring the need for larger, more balanced datasets and external validation efforts.

Our results are consistent with those reported by Olshvang et al. [54], who applied RF, LR, LassoNet, and XGBoost models to predict lean mass based on anthropometric and sociodemographic data (age, ethnicity, body mass, stature, and waist girth) using data from the National Health and Nutrition Examination Survey (NHANES). In their study, body mass and male sex were among the most important predictors, and they concluded that RF, XGBoost, and LassoNet accurately predicted both total and appendicular lean masses. Similarly, Cichosz et al. [55] used NHANES data and ANN to predict ALMI and fat mass in adults, reporting strong correlations with DXA measurements. Buccheri et al. [56] also used NHANES data to develop a near-zero-cost screening tool using DT to emulate DXA performance for identifying low muscle mass. They found that anthropometric measurements of the lower limbs provided a simpler yet effective alternative, with an AUC of 0.88–0.90.

Marazzato et al. [57] demonstrated the feasibility of predicting ALMI via DXA using a recurrent neural network (RNN) in a diverse population consisting of 576 children, adolescents, and adults. The model included 10 demographic dimensions (sex, age, and seven ethnic groups) and 43 anthropometric dimensions obtained from a 3D optical scanner. The predicted and measured ALMI values were highly correlated, with small mean, absolute, and squared prediction errors, highlighting the potential of ANN in body composition prediction using large-scale digital anthropometric data. In India, Birk et al. [58] developed a machine learning model to estimate body composition using bioimpedance, skinfolds, body girths, and grip strength. Their model indicated lower prediction errors than traditional equations. This is consistent with the study by Guo et al. [59], who developed an online calculator using ML to predict low ALMI. Their model, based on XGBoost and using only stature, waist girth, age, and race, achieved an AUC above 0.85 in validation tests. The tool was designed for community-level use in the U.S., facilitating early sarcopenia detection and intervention.

In a follow-up study, Buccheri et al. [60] developed computationally simple equations to estimate DXA-measured ALMI using 38 non-laboratory variables in older adults. Using only body mass, sex, and anthropometric measures (thigh and arm girth), they achieved an AUC-ROC of 0.89. Interestingly, they found that adding more than three variables did not improve model performance. Shi et al. [61] developed an anthropometric equation using LASSO regression to estimate ALMI in elderly women in India. The model included body mass, stature, BMI, sitting stature, waist-to-hip ratio (WHR), upper arm length, and other limb length summaries. The final equation—based on body mass, WHR, upper arm length, and sitting stature—demonstrated good agreement with DXA, with 95% limits of agreement. This is consistent with the findings of our study, in which BMI emerged as an important predictor of lean mass in women. Similarly, Kang et al. [62] also identified BMI as a key variable for predicting muscle mass in women and reported the highest accuracy using boosted algorithms. Multiple recent studies have focused on predicting sarcopenia, pre-sarcopenia, or similar conditions using accessible data sources, yielding promising and practical results, showing the importance of continuing the research in this field [63,64,65,66,67]. It is worth noting that most of these studies focused on predicting ALMI as a continuous variable (in kilograms) rather than its categorical classification (normal vs. low). In contrast, we employed a categorical prediction approach to avoid the limitations associated with defining population-specific cut-off points for sarcopenia diagnosis. As highlighted by Rangel-Peniche et al. [26], these cut-offs may vary across ethnicities and populations. Predicting ALMI status using anthropometric data may thus offer practical advantages in the clinical screening and diagnosis of sarcopenia.

As shown, the heterogeneity in model types and anthropometric variables across studies makes direct comparisons difficult. Nevertheless, ML tools are valuable alternatives when DXA is unavailable, potentially offering scalable solutions for sarcopenia screening via mobile applications or clinical software. Predicting ALMI status categorically is advantageous, as it directly informs the diagnostic component of sarcopenia and reduces variability and bias due to different cut-off definitions. The variability in feature importance across sex groups (BMI and CCG in women and ARG in men) further suggests that sex-specific models may be more appropriate than unisex models. These differences may be due to distinct patterns of fat and muscle distribution between men and women and merit further study in larger and more diverse samples [68,69].

One of the main limitations of this study was the sample size, particularly in the male group. This issue is not uncommon in the literature, as large-scale studies with comprehensive assessments are often constrained by high costs and limited resources [70,71]. Determining an appropriate sample size for predictive models developed using ML techniques is not straightforward [72]. Several studies have reported small sample sizes (with a median of 88 participants), and, in certain cases, higher accuracy has been observed in models trained on smaller datasets [71]. For example, Castillo-Olea et al. [73] used 166 participants in Tijuana, Mexico; Pineda-Zuluaga et al. [74] worked with 237 in Manizales, Colombia; and Abdalla et al. [75] studied 125 individuals in São Paulo, Brazil. Some datasets are vulnerable when the ratio of features to sample size is high, increasing the likelihood that the model fits noise rather than the underlying data patterns. Consequently, ML models may produce overly optimistic results when trained on small datasets, resulting in poor generalizability [71,76]. Although small samples are more prone to overfitting and less likely to detect true effects, high-quality data may compensate for this limitation [70]. Specifically, for neural networks, it has been recommended that the sample size be at least 50 times the number of model weights to ensure robustness and reduce bias [76].

Rajput et al. [70] found that predictive error decreased with sample sizes above 120, and optimal performance was achieved with more than 1000 participants. In our study, the female group met the minimum threshold for reliable modeling, while the smaller male sample likely limited model stability. Notably, once the minimum sample size is reached, increasing the number of participants does not significantly improve model performance, thus offering a favorable cost–benefit ratio [70].

Furthermore, external validation is essential to confirm the generalizability of ML models that demonstrate potential clinical value [56]. In this study, internal validation was performed by dividing the dataset into training and testing sets, which helps minimize optimistic bias. Nevertheless, even with reasonable sample sizes, this approach may be insufficient. The next step should involve validation using an independent sample, ideally from a different region or population subgroup. Cross-validation remains a widely used approach when available data are limited [70].

Another limitation is that the use of anthropometric measurements is prone to various sources of error, including lack of equipment calibration, data recording inaccuracies, and insufficient training or expertise of the anthropometrist. Even environmental conditions, such as room temperature, can influence measurement outcomes. In addition, skinfold and girth measurements can be affected by physiological factors (e.g., hydration status, fat redistribution in aging) and inter-observer variability, limiting precision. Despite these challenges, the use of standardized protocols, such as those established by ISAK, and proper certification can substantially reduce both intra- and inter-observer measurement error [39,77,78]. These considerations are particularly important in older adults, where physiological changes and greater variability in body composition increase the need for precise and reliable assessments [79]. In the context of clinical applicability, ensuring reproducibility and precision is essential, especially when working with diverse populations. This further underscores the importance of adopting validated, standardized methods tailored to specific populations, such as older Mexican adults, to support accurate diagnosis and appropriate clinical decision-making.

Despite these limitations, the models still achieved high accuracy, emphasizing the value of continuing to expand the dataset to improve predictive performance. Future research should consider adding new predictive variables or exploring data augmentation techniques such as SMOTE (Synthetic Minority Oversampling Technique) to handle imbalanced class distributions [80]. The clinical applicability of these models lies in their potential to be implemented in digital platforms or mobile applications for use by trained health professionals. Such tools could provide automated assessments of ALMI levels or flag individuals at risk of sarcopenia. For nationwide implementation, however, these tools must undergo rigorous external validation and standardization across population subgroups. This study represents a significant step toward developing anthropometry-based sarcopenia screening tools tailored to the Mexican older adult population.

Additionally, the lack of a global consensus on sarcopenia definitions and diagnostic criteria remains a challenge [81]. This uncertainty hinders comparability across studies and emphasizes the need for accessible, replicable tools based on interpretable and reliable predictors. Accurate evaluation of body composition using accessible methods is crucial for understanding both the health and disease of an older person, as variations in muscle and fat distribution are associated with conditions such as sarcopenia, obesity, and metabolic disorders. Conventional approaches, such as DXA and bioelectrical impedance analysis (BIA), have limited capacity to comprehensively and efficiently assess body composition in clinical settings. AI, particularly through ML, enhances the segmentation and analysis of these techniques, offering greater precision and accuracy. This enables more personalized healthcare by identifying specific patterns of muscle and fat distribution associated with disease risk [82].

## 5. Conclusions

There is an urgent need for non-invasive, cost-effective, and scalable methods for population-level screening while also providing practical support to clinicians. Anthropometry stands out as a valuable tool for identifying low ALMI in older adults with sarcopenia. Additionally, ML algorithms are gaining prominence, demonstrating strong predictive performance [83]. As highlighted in prior research, ML is becoming an increasingly powerful asset in healthcare, especially for the early detection of sarcopenia in older populations. These techniques excel at analyzing large and complex datasets, facilitating timely diagnosis and intervention.

Continued research is essential to refine and validate diagnostic tools for sarcopenia, including integrating emerging technologies such as wearable activity trackers and smartwatches. A recent comprehensive review highlights that these devices can offer valuable insights into sarcopenia progression, support monitoring, and indicate the need for early intervention [81]. AI and ML offer promising solutions to global healthcare challenges by driving innovation, improving efficiency, and expanding access to care. Moreover, they have the potential to revolutionize healthcare systems through enhanced disease prediction, earlier diagnosis, personalized treatment strategies, and improved equity. In this context, the clinical applicability of the models developed in this study will depend on the creation of a user-friendly platform—possibly in the form of a mobile or desktop application—for use by trained healthcare professionals. Such tools could estimate ALMI levels or predict sarcopenia diagnosis based on simple anthropometric inputs. Although further research and external validation are needed for national-level implementation, this study represents an important step toward improving sarcopenia detection in the older Mexican adult population. Moreover, this work adds to the growing evidence that AI-supported anthropometric assessments can help bridge current diagnostic gaps and improve health outcomes in aging populations. However, successful implementation depends on responsible and ethical governance, integration into healthcare infrastructure, secure data management, and robust user engagement [32,82]. Ethical reviews emphasize the need to address privacy, transparency, bias, and user autonomy when applying AI in clinical contexts [83]. Challenges remain, including ensuring input data accuracy and encouraging user adherence, which must be addressed to fully realize the benefits and potential of these technologies [81].

## Figures and Tables

**Figure 1 jfmk-10-00276-f001:**
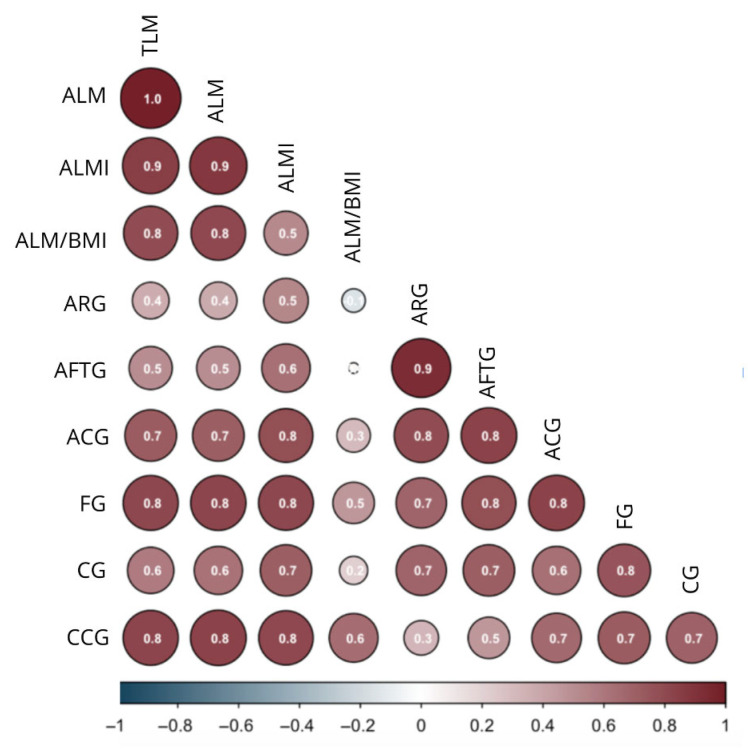
Correlation diagram between anthropometric and body composition variables of the older persons from the SAMJ study. ALM: Appendicular Lean Mass; ALMI: Appendicular Lean Mass Index; BMI: Body Mass Index; ARG: Arm Relaxed Girth; AFTG: Arm Flexed and Tensed Girth; ACG: Arm Corrected Girth; FG: Forearm Girth; CG: Calf Girth; CCG: Calf Corrected Girth. Pearson or Spearman correlation, depending on data normality. The numbers represent the correlation coefficient (r).

**Figure 2 jfmk-10-00276-f002:**
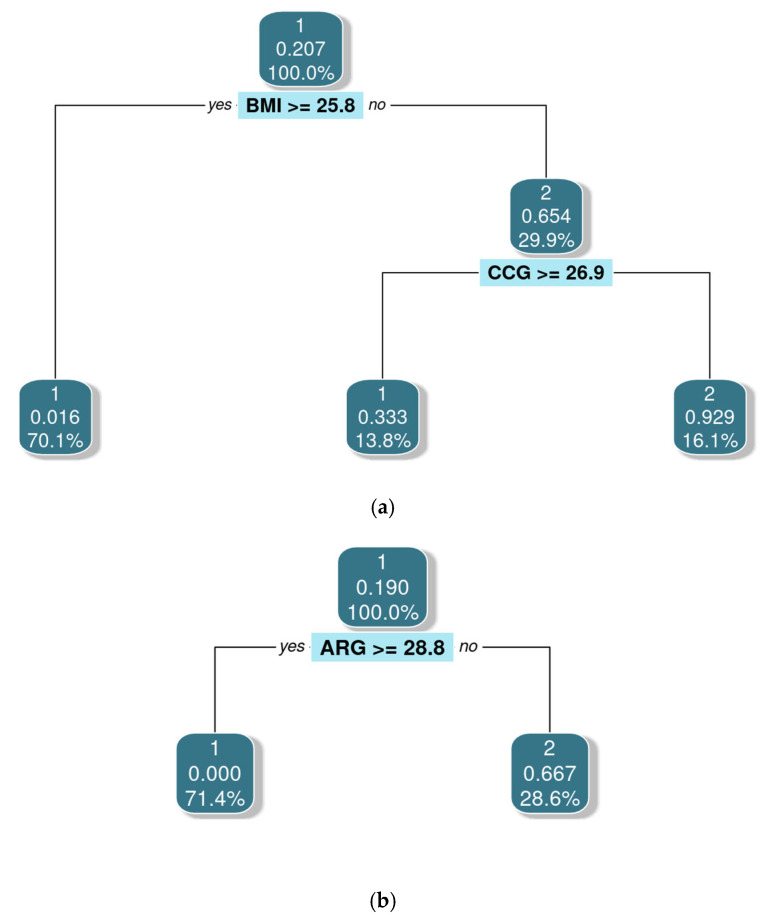
Decision Tree for Appendicular Lean Mass Index by sex in older adults from the SAMJ study. (**a**) Females. (**b**) Males. 1 = Normal ALMI; 2 = Low ALMI; Number of trees = 87; BMI: Body Mass Index; CCG: Calf Corrected Girth; ARG: Arm Relaxed Girth.

**Figure 3 jfmk-10-00276-f003:**
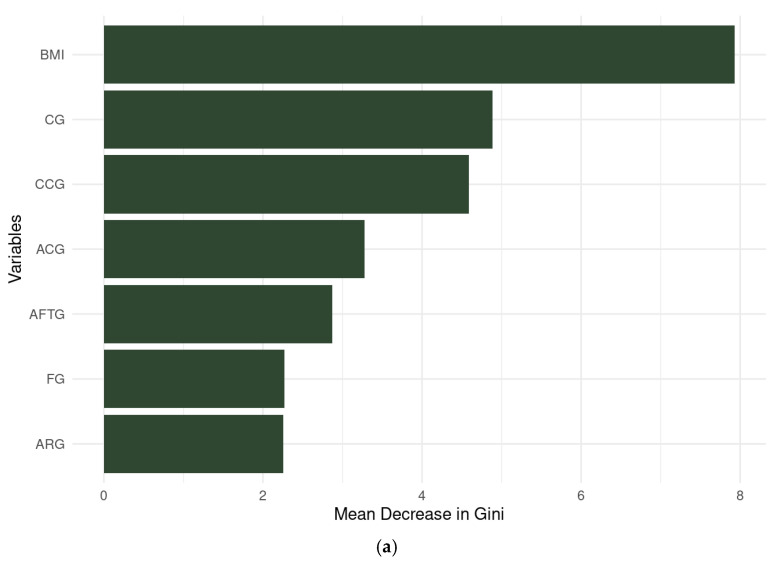
Variables with major significance for ALMI level in Random Forests. (**a**) Females, (**b**) Males. BMI: Body Mass Index; ARG: Arm Relaxed Girth; AFTG: Arm Flexed and Tensed Girth; FG: Forearm Girth; CG: Calf Girth; ACG: Arm Corrected Girth; CCG: Calf Corrected Girth. Mean Decrease in Gini: It is a measure of node purity used to construct Decision Trees, representing how mixed the classes are in a node. High values indicate that the variable contributes significantly to improving node purity, which implies that it is an important variable for predicting the response or classifying correctly.

**Figure 4 jfmk-10-00276-f004:**
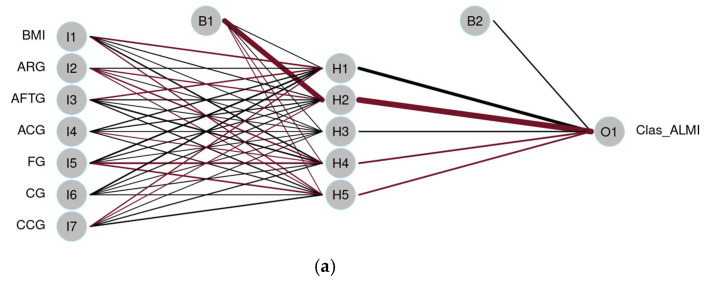
Artificial Neural Network for the prediction of ALMI level. (**a**) Females. (**b**) Males. BMI: Body Mass Index; ARG: Arm Relaxed Girth; AFTG: Arm Flexed and Tensed Girth; FG: Forearm Girth; CG: Calf Girth; ACG: Arm Corrected Girth; CCG: Calf Corrected Girth; 1: normal; 2: low. Line color denotes the direction of the weight: black for positive and burgundy for negative weights.

**Figure 5 jfmk-10-00276-f005:**
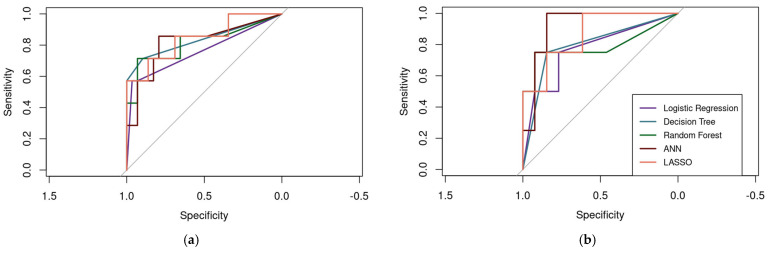
ROC curves of the machine learning models to predict ALMI in older adults from the SAMJ study. (**a**) Females. (**b**) Males.

**Table 1 jfmk-10-00276-t001:** Sex differences in quantitative body composition variables with normal distribution in older adults from the SAMJ study.

	Male Mean (SD)	Female Mean (SD)	*p*-Value
Body composition			
ALMI (kg/m^2^)	8.13 (0.94)	6.66 (0.85)	<0.001
Anthropometry			
BMI (kg/m^2^)	28.6 (3.85)	28.99 (5.05)	0.542
ARG (cm)	30.94 (3.24)	30.84 (4.05)	<0.001
AFTG (cm)	31.62 (3.03)	30.04 (3.98)	<0.001
FG (cm)	27.03 (1.99)	24.08 (2.12)	<0.001
CG (cm)	36.27 (2.9)	34.59 (3.47)	<0.001
CoG arm (cm)	26.94 (2.35)	24.07 (2.64)	<0.001
CoG calf (cm)	36.27 (2.55)	28.06 (2.5)	<0.001

ALMI: Appendicular Lean Mass Index; BMI: Body Mass Index; ARG: Arm Relaxed Girth; AFTG: Arm Flexed and Tensed Girth; FG: Forearm Girth; CG: Calf Girth; CoG: Corrected Girth. *p*-values are from Student’s *t*-tests.

**Table 2 jfmk-10-00276-t002:** Sex differences in quantitative body composition variables without normal distribution in older adults from the SAMJ study.

Body Composition	Male Median (IQR)	Female Median (IQR)	*p*-Value
ALM (kg)	22.13 (3.7)	11.68 (3.37)	<0.001
ALM/BMI	0.71 (0.13)	0.53 (0.1)	<0.001

ALM: Appendicular Lean Mass; BMI: Body Mass Index; IQR: Interquartile Range. *p*-values are from the Mann–Whitney U test, with statistically significant results in bold.

**Table 3 jfmk-10-00276-t003:** Cut-off points for sarcopenia diagnosis in older adults from the SAMJ study.

Sex	ALMI (kg/m^2^)	ALM/BMI
Male	<7.49	<0.69
Female	<5.93	<0.46

ALMI: Appendicular Lean Mass Index; ALM: Appendicular Lean Mass; BMI: Body Mass Index; Cut-off points were determined using the 20th percentile.

**Table 4 jfmk-10-00276-t004:** Correlation matrix between anthropometric and DXA variables of the older persons from the SAMJ study.

Anthropometry	DXA Fat Mass (kg)			DXA Lean Mass (kg)				
	Appendicular r	Arms r	Legs r	Appendicular r	Arms (kg) r	Legs (kg) r	ALMI (kg/m^2^) r	ALM/BMI r
Body mass (kg)	0.496 **	0.474 **	0.479 **	0.802 **	0.744 **	0.809 **	0.798 **	0.317 **
BMI (kg/m^2^)	0.731 **	0.702 **	0.704 **	0.731 **	0.287 **	0.352 **	0.589 **	−0.287 **
TSF (mm)	0.772 **	0.754 **	0.740 **	−0.235 **	−0.283 **	−0.209 *	−0.065	−0.607 **
CSF (mm)	0.721 **	0.613 **	0.718 **	−0.299 **	−0.373 **	−0.269 **	−0.159 *	−0.579 **
ARG (cm)	0.698 **	0.733 **	0.641 **	0.400 **	0.400 **	0.396 **	0.547	−0.137 **
AFTG (cm)	0.574 **	0.654 **	0.522 **	0.527 **	0.529 **	0.515 **	0.646 **	0.037
FG (cm)	0.303 **	0.359 **	0.271 **	0.825 **	0.824 **	0.818 **	0.809 **	0.461**
CG (cm)	0.549 **	0.444 **	0.554 **	0.630 **	0.540 **	0.655 **	0.701 **	0.202 *
Calf CoG (cm)	−0.562	−0.080	−0.049	0.831 **	0.805 **	0.825 **	0.786 **	0.650 **
Arm CoG (cm)	0.317 **	0.430 **	0.267 **	0.710 **	0.685 **	0.773 **	0.293 **	0.202 *

ALMI: Appendicular Lean Mass Index; ALM: Appendicular Lean Mass; BMI: Body Mass Index; TSF: Triceps Skinfold; CSF: Calf Skinfold; ARG: Arm Relaxed Girth; AFTG: Arm Flexed and Tensed Girth; FG: Forearm Girth; CG: Calf Girth; CoG: Corrected Girth; DXA: Dual-energy X-ray Absorptiometry; r = correlation coefficient. Pearson or Spearman correlation, depending on data normality, *****
*p* < 0.05, ******
*p* < 0.001.

**Table 5 jfmk-10-00276-t005:** Logistic Regression model for predicting ALMI level using anthropometric variables stratified by sex among older persons from the SAMJ study.

	Estimate		Standard Error		Z Value		*p* Value	
	F	M	F	M	F	M	F	M
Intercept	7376.80	1802.22	706,862.64	973,842.13	0.010	0.002	0.992	0.999
BMI	−185.10	−3.13	15,623.58	14,513.51	−0.012	0.000	0.991	1.000
ARG	506.38	−56.27	40,247.17	99,777.97	0.013	−0.001	0.990	1.000
AFTG	−331.97	24.36	28,252.76	44,860.02	−0.012	0.001	0.991	1.000
Arm CoG	−222.99	23.28	19,574.73	45,842.11	−0.011	0.001	0.991	1.000
FG	−48.82	−21.73	6982.93	33,221.02	−0.007	−0.001	0.994	0.999
CG	−48.11	19.88	10,491.04	26,609.56	−0.005	0.001	0.996	0.999
Calf CoG	−14.11	−47.95	3338.89	29,310.10	−0.004	−0.002	0.997	0.999

BMI: Body Mass Index; ARG: Arm Relaxed Girth; AFTG: Arm Flexed and Tensed Girth; FG: Forearm Girth; CG: Calf Girth; CoG: Corrected Girth; F: Female; M: Male.

## Data Availability

The data presented in this study are available on request from the corresponding authors due to agreements ensuring participant confidentiality. The full R Markdown analysis, including all data processing, modeling steps, and results, is publicly accessible at: https://rpubs.com/anglez02/1317079 (accessed on 14 July 2025).

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
