# Peer review of "Anthropometric Measurements for Predicting Low Appendicular Lean Mass Index for the Diagnosis of Sarcopenia: A Machine Learning Model"

_jfmk, 2025, doi:10.3390/jfmk10030276_

Round 1

Reviewer 1 Report

Comments and Suggestions for Authors

The manuscript entitled "Anthropometric Measurements to Predict Low Appendicular Lean Mass Index for the Diagnosis of Sarcopenia: A Machine Learning Model" has been reviewed. This study explores the use of machine learning models to predict low appendicular lean mass index as a proxy for sarcopenia diagnosis using accessible anthropometric measurements in Mexican older adults. The methodology of this study is appropriate, and the manuscript is overall clearly written. 

The relatively small sample size and sex imbalance may limit generalizability. The discussion should expand on how it may affect the stability of the models, particularly for male participants.

The manuscript would benefit from a more detailed discussion on external validation.

Given the clinical audience, more emphasis could be placed on how the decision trees and feature importance in Random Forests enhance interpretability. 

Comments on the Quality of English Language

A language polish to ensure consistency in tense and minor grammatical corrections would be beneficial

Author Response

Dear Reviewer,

Please find the attached document for your review.

Reviewer 2 Report

Comments and Suggestions for Authors

Dear Sirs, I find your study interesting. I'll leave some comments aimed at improving what you present.
Regards.

Moreover, the practical utility of DXA for muscle mass estimation in clinical practice has also been questioned [12].
I think it's necessary to specify in detail what is being questioned.

Nevertheless, further research is needed to evaluate and compare different lean mass estimation methods tailored to specific ethnic groups [11].

What topics need to be investigated? Add that information.

In this context, limited research has been conducted in the Mexican population. Therefore, additional studies are necessary to evaluate and compare various muscle mass assessment methods, tailored to the specific characteristics of this population.

Why is it limited? What things need to be investigated? What do the studies already conducted on the topic say?
I think it's necessary to add more information to justify the study.

I think it's necessary to add some more information about the data collection process. Were the subjects receiving assistance? At what time in the morning? Were they all assessed at the same time? What clothes were they wearing? Everything necessary to accurately understand the characteristics of the people assessed, and the conditions they were in.
Also, more details about the location where the assessments were taken.

I think the discussion is good, as it compares their results with other models, highlighting their similarities and differences. In some cases, these are good and in others, not so good. But the important thing is that it highlights the advantages and limitations.

I think that before the limitations and advantages, it is necessary to highlight the importance of what they propose for practice and the contribution they make to science (something is discussed elsewhere, but I think it needs to be emphasized even more).

The limitations of the study should include the "limitations" of measuring anthropometric variables (fold thicknesses, perimeters, diameters).

Final question: Can anyone access what they propose? It would be interesting if at least the people who read your research could access and use the tool, perhaps not with the same population, but it would be helpful.

I hope my comments contribute to improving your research.
Congratulations on your work.

Author Response

(The authors gave the same response as above.)
